# Impact of the COVID-19 pandemic on exercise habits and overweight status in Japan: A nation-wide panel survey

**Sae Ochi[1]\*, So Mirai[2], Sora Hashimoto[3], Yuki Hashimoto[4], Yoichi Sekizawa[4]**

**1** Department of Laboratory Medicine, The Jikei University School of Medicine, Tokyo, Japan, **2** Department of Psychiatry, Tokyo Dental College, Tokyo, Japan, **3** United Health Communication Co. Ltd., Tokyo, Japan, **4** Research Department, Research Institute of Economy, Trade and Industry, Tokyo, Japan

\* ochisae1024@jikei.ac.jp

## Abstract

A catastrophic disaster may cause distant health impacts like immobility and obesity. The aim of this research was to analyse the association of the COVID-19 pandemic and lifestyle factors -exercise habit and overweight status in the Japanese population. Nation-wide online questionnaires were conducted five times from October 2020 to October 2021. The changes in exercise habit, body mass index (BMI) and overweight status (BMI >25kg/m$^2$) were compared between the first questionnaire and a questionnaire conducted a year later. Risk factors for losing exercise habit or becoming overweight were analysed using multiple regression. Data were obtained from 16,642 participants. In the early phase of the pandemic, people with high income and elderly females showed a higher risk for decreased exercise days. The proportion of overweight status increased from 22.2% to 26.6% in males and from 9.3% to 10.8% in females. Middle-aged males, elderly females, and males who experienced SARS-CoV-2 infection were at higher risk of becoming overweight. Our findings suggest that risks for immobility and overweight are homogeneous. Continuous intervention for elderly females and long-term intervention for males infected with SARS-CoV-2 might be especially needed. As most disasters can cause similar social transformation, research and evaluation of immobility and obesity should address future disaster preparation/mitigation plans.

## Introduction

During and after a catastrophic disaster population health may deteriorate in many ways. This impact on health is not limited to direct acute conditions such as injuries, but also includes indirect and chronic effects caused by lifestyle changes, mental stress, job losses, and social disruption. In particular, after chemical, biological, radiological, nuclear, or explosive (CBRNE) disasters, fear about invisible hazards may cause social panic that often leads to a deterioration in the health of the population. For example, after the Fukushima Daiichi nuclear power plant accident in 2011, the limitation of outdoor activities from fear of radiation exposure and other

**Data Availability Statement:** The data used in the present study belong to the RIETI and can be obtained from the institute upon reasonable request.

**Funding:** This work was supported by the Project Grant from the Co-creation Center for Disaster Resilience, IRIDeS, Tohoku University to SO. The funders had no role in study design, data collection and analysis, decision to publish, or preparation of the manuscript.

**Competing interests:** The authors have declared that no competing interests exist.

lifestyle changes led to an increase in metabolic syndromes such as hyperlipidaemia [1] and diabetes mellitus [2]. Some researchers estimated that this increase may have even shortened life expectancies to a greater extent than the small amount of radiation exposure caused by the accident [3]. Other health impacts among the evacuees included a decline in physical performance [4], increased obesity [5, 6], and a deterioration in mental status [7]. As the size of an indirect health impact surpasses that of a direct impact, preventing the indirect impacts is a key to retaining health in disaster areas.

Another type of CBRNE disaster, biological disaster, is a disaster caused by the rapid spread of disease caused by microorganisms. As fear of invisible microorganisms can cause social panic, the situation similar to nuclear disaster may happen. However, there is a paucity of research on these indirect health impacts and therefore more research is needed to address chronic conditions after a disaster and how communities can prepare and respond to disasters and public health emergencies.

The SARS-CoV-2 pandemic that started in late 2019 is one of the largest biological disasters of this decade. The virus had killed more than six million people by the end of June 2022 [8]. In addition, nationwide lockdowns, policies to encourage social distancing, travel restrictions, and voluntary bans of many activities in many countries may have caused severe social disruption and led to lifestyle changes such as alterations in eating habits [9, 10] and a decline in physical activities [11].

As a consequence of these changes, experts have raised concerns about an increase in the prevalence of obesity during and after the pandemic [12, 13]. Furthermore, previous research has suggested that COVID-19 itself may increase the risk of obesity [14]. However, the effect of such social disruption may be heterogeneous. A previous study targeting the population with obesity showed that only a limited number of people were vulnerable to lifestyle changes [15]. Other online surveys have even reported an improvement in body mass index (BMI) and eating habits among some groups of people [16, 17]. However, as these studies targeted the relatively younger population, there is a limitation in the generalizability of the findings. Therefore, a nation-wide survey is needed to understand the size and nature of the indirect impacts of the pandemic on risk factors associated with adverse health outcomes.

The "Continuing survey on mental and physical health during the COVID-19 pandemic" is a nationwide, longitudinal, online survey carried out by the Research Institute of Economy, Trade and Industry, Japan (RIETI), Japan. The current study used this data to analyse time trends and risk factors for exercise habit and obesity in addition to attitudes regarding vaccination [18] and infection avoidance behaviour in Japan [19]. The results will provide additional insight on the health impact of the pandemic and therefore will provide clues for developing effective disaster mitigation plans for future CBRNE disasters.

## Materials and methods

The detailed method for data collection is described in our previous reports [18, 19]. In short, nation-wide online questionnaires were conducted five times: October 2020, and January, April, July, and October 2021. The questionnaire was conducted only for a year mainly due to limited finance. The online survey was called "the 2020 Continuing survey on mental and physical health during the COVID-19 pandemic" (hereinafter named the RIETI questionnaire survey), with the NTTCom Online Marketing Solutions Corporation commissioned to conduct and anonymize the survey. The researchers were provided with only de-identified data.

## Target population

The participants were Japanese people aged 18–74 years living in Japan who were randomly selected from the database of registered monitors of the NTTCom. The participants were selected so that the demographic composition ratios of sex, age, and distribution of residential prefectures matched the population estimates of the Statistics Bureau of Japan (final estimates, May 2020). The aim was to enrol 20000 participants according to the eligibility of our study fund.

## Data collected

The following data were collected

- Background information: sex, age group, pre-existing conditions, marital status, yearly income, height, weight, and exercise habit before the COVID-19 pandemic

- Infection status of SARS-CoV-2: past diagnosis, current infection, or no infection

- Activities to avoid the virus: avoid poorly ventilated places, avoid crowded places, wear a mask, wash hands, disinfect belongings, gargle, change clothes frequently, keep a distance from others, refrain from seeing a doctor, and refrain from going out as much as possible

- Exercise habit: days of exercise per week

- Health status: patient health questionnaire 9 (PHQ-9) for depression status [20], GAD-7 for anxiety [21], and subjective health status on a six-point scale

- Change in economic status compared to the previous questionnaire

## Exclusion criteria

As the online survey was written in Japanese, people who could not read Japanese were excluded. After collection, the data were excluded for individuals who provided seemingly inappropriate answers, including non-existent zip codes, extreme outlying values for height and weight, and controversial answers throughout the five questionnaires such as a difference in age of two years or more. The respondents who took a very short time (less than five minutes) or a very long time (ten hours or longer) to answer the survey questions were also excluded.

## Definition of changes in the early and late phases

We defined the period of the first and second questionnaire as the 'early phase' and that of the fifth questionnaire as the 'late phase' of the pandemic. Changes in habits in the early phase were evaluated by comparing the answers in the first and second questionnaires, while changes in the late phase were evaluated by comparing the answers in the first and fifth questionnaires.

## Definition of exercise habit, obesity, and overweight

People who answered that they did not exercise (i.e., 0 per week) were categorised as 'no exercise habit'. Changes in exercise habit were estimated by calculating the difference in exercise days at the time of each questionnaire, compared to that stated in the first questionnaire.

Obesity and overweight were defined as a BMI $>30$ kg/m$^2$ and $>25$ kg/m$^2$, respectively. As the proportion of obesity is not high in the Japanese population, the proportion of overweight status was used as an outcome for further analysis. Newly developed overweight status was

defined as those who were not overweight at the time of the first questionnaire but became overweight in the following periods.

## Statistical analysis

A change in exercise habit during the early phase was calculated by subtracting the exercise days per week in the second questionnaire from the days in the first questionnaire. A change during the late phase was calculated by subtracting the exercise days per week at the time of the fifth questionnaire from the days in the first questionnaire. The difference between exercise days in the first questionnaire and the following questionnaires were analysed using the paired t-test.

The social and psychological impact of the pandemic in Japan has been reported to be different according to sex [22, 23]. Therefore, the statistical analyses were separately conducted by sex. Differences between males and females were compared using the chi-square test.

Factors associated with changes in exercise days per week and risk factors for developing overweight status were analysed using a multiple linear regression model. For the sensitivity analysis, the analysis was conducted using factors in the early and later phases.

The statistical analyses were carried out using Stata/SE 16.0 (StataCorp LLC, College Station, TX, USA). *P*-values of < 0.05 were considered to be statistically significant.

## Ethical consideration

Written consent for participation was obtained online from all individuals who participated in the study. The present study was conducted with the approval of the ethics committee of Hiramatsu Memorial Hospital (No: 20200925).

## Results

Of the 19,340 participants, 2,698 were excluded due to providing inappropriate or controversial answers. The remaining 16,642 (8022 males and 8,620 females) were included in the final analysis. The background of the participants grouped by sex is shown in Table 1.

According to the National Health and Nutrition Survey in Japan 2019 [24], about 33% of males and 29% of females had exercise habit of ≥ 2 days per week. Our data showed slightly higher percentage (36% in male and 34% in female) answered they had ≥ 2 days per week of exercise habit.

There was a significant difference between sexes in all the variables except for the prevalence of cancer. Female participants were more likely to take any infection avoidance behaviour. Male participants had a higher prevalence of exercise habit than females. During the survey, 5,117 (30.7%) of the participants provided at least one missing data or controversial response in the late phase.

Proportion of missing data in each questionnaire by sex and age category is shown in Table 2. The proportions increase with time and were higher among female participants and participants at younger age (≤30 years old).

The comparison of the background of those with missing data and those with complete data are compared in the S1 Table. Younger people, those with lower income levels, and those who had never married were more likely to provide essential data.

### Changes in exercise habit

The changes in exercise habits at the time of each questionnaire are shown in Fig 1. The proportion of people who reported less exercise days per week than that at baseline (October 2020) increased gradually with time, while those who reported more exercise days at baseline

**Table 1. Background of the participants.** Differences between males and females were calculated by the chi-squared test for categorical variables and the t-test for numerical variables.

| | | Total (N = 16,642) | | Male (N = 8,022) | | Female (N = 8,620) | | P |
|---|---|---|---|---|---|---|---|---|
| **Categorical variables** | | N | % | N | % | N | % | |
| Age group (yr) | < = 30 | 2,813 | 16.9 | 1,081 | 13.5 | 1,732 | 20.1 | <0.01[†] |
| | 30–39 | 2,053 | 12.3 | 939 | 11.7 | 1,114 | 12.9 | |
| | 40–49 | 3,256 | 19.6 | 1,609 | 20.1 | 1,647 | 19.1 | |
| | 50–59 | 3,316 | 19.9 | 1,750 | 21.8 | 1,566 | 18.2 | |
| | 60–69 | 3,580 | 21.5 | 1,783 | 22.2 | 1,797 | 20.8 | |
| | 70–74 | 1,624 | 9.8 | 860 | 10.7 | 764 | 8.9 | |
| Income (10,000yen/year)* | <300 | 4,292 | 25.8 | 1,873 | 23.3 | 2,419 | 28.1 | <0.01[†] |
| | 300–500 | 4,589 | 27.6 | 2,226 | 27.7 | 2,363 | 27.4 | |
| | 500–700 | 3,179 | 19.1 | 1,518 | 18.9 | 1,661 | 19.3 | |
| | 700–1000 | 2,795 | 16.8 | 1,428 | 17.8 | 1,367 | 15.9 | |
| | >1000 | 1,787 | 10.7 | 977 | 12.2 | 810 | 9.4 | <0.01[†] |
| Marital status | Married | 9,905 | 59.5 | 4,814 | 60.0 | 5,091 | 59.1 | <0.01[†] |
| | Divorced | 949 | 5.7 | 362 | 4.5 | 587 | 6.8 | |
| | Widow/widower | 366 | 2.2 | 88 | 1.1 | 278 | 3.2 | |
| | Never married | 5,422 | 32.6 | 2,758 | 34.4 | 2,664 | 30.9 | |
| Pre-existing condition | Obesity | 567 | 3.4 | 364 | 4.5 | 203 | 2.3 | <0.01 |
| | Overweight | 2,586 | 15.5 | 1,783 | 22.2 | 803 | 9.3 | <0.01[†] |
| | Hypertension | 2,529 | 15.2 | 1,717 | 21.4 | 812 | 9.4 | <0.01[†] |
| | Dyslipidemia | 1,463 | 8.8 | 797 | 9.9 | 666 | 7.7 | <0.01[†] |
| | Diabetes | 796 | 4.8 | 610 | 7.6 | 186 | 2.2 | <0.01[†] |
| | Heart disease | 378 | 2.3 | 258 | 3.2 | 120 | 1.4 | <0.01[†] |
| | Lung or respiratory disease | 376 | 2.3 | 183 | 2.3 | 193 | 2.2 | <0.01[†] |
| | Renal disease | 120 | 0.7 | 76 | 0.9 | 44 | 0.5 | <0.01[†] |
| | Cancer | 242 | 1.5 | 108 | 1.3 | 134 | 1.6 | 0.26 |
| | Other condition¶ | 235 | 1.4 | 94 | 1.2 | 141 | 1.6 | 0.01[†] |
| Activities to avoid virus | Avoid poorly ventilated places | 14,167 | 85.1 | 6,458 | 80.5 | 7,709 | 89.4 | <0.01[†] |
| | Avoid crowded places | 14,433 | 86.7 | 6,743 | 84.1 | 7,690 | 89.2 | <0.01[†] |
| | Avoid talking at close distances | 13,264 | 79.7 | 6,213 | 77.4 | 7,051 | 81.8 | <0.01[†] |
| | Wear a mask | 13,264 | 79.7 | 6,213 | 77.4 | 7,051 | 81.8 | <0.01[†] |
| | Wash hands | 16,019 | 96.3 | 7,574 | 94.4 | 8,445 | 98.0 | <0.01[†] |
| | Change clothes frequently | 3,579 | 21.5 | 1,583 | 19.7 | 1,996 | 23.2 | <0.01[†] |
| | Gargle | 11,433 | 68.7 | 5,245 | 65.4 | 6,188 | 71.8 | <0.01[†] |
| | Disinfect belongings | 4,859 | 29.2 | 1,986 | 24.8 | 2,873 | 33.3 | <0.01[†] |
| | Keep distance from others | 13,623 | 81.9 | 6,274 | 78.2 | 7,349 | 85.3 | <0.01[†] |
| | Refrain from seeing a doctor | 8,351 | 50.2 | 3,664 | 45.7 | 4,687 | 54.4 | <0.01[†] |
| | Refrain from going out | 10,151 | 61.0 | 4,606 | 57.4 | 5,545 | 64.3 | <0.01[†] |
| | Exercise regularly | 10,084 | 60.6 | 5,039 | 62.8 | 5,045 | 58.5 | <0.01[†] |
| Exercise habit (days /week) | 0 | 6,558 | 39.4 | 2,983 | 37.2 | 3575 | 41.5 | <0.01[†] |
| | 1 | 2,454 | 14.7 | 1,261 | 15.7 | 1193 | 13.8 | |
| | 2 | 1,844 | 11.1 | 939 | 11.7 | 905 | 10.5 | |
| | 3 | 1,494 | 9.0 | 649 | 8.1 | 845 | 9.8 | |
| | ≥ 4 | 4,292 | 25.8 | 2,190 | 27.3 | 2102 | 24.4 | |
| Income change | No change | 10,941 | 65.7 | 5,321 | 66.3 | 5,620 | 65.2 | <0.01[†] |
| | Increase | 4,852 | 29.2 | 2,274 | 28.3 | 2,578 | 29.9 | |
| | Decrease | 849 | 5.1 | 427 | 5.3 | 422 | 4.9 | |

*(Continued)*

**Table 1.** (Continued)

| Categorical variables | | Total (N = 16,642) | | Male (N = 8,022) | | Female (N = 8,620) | | *P* |
|---|---|---|---|---|---|---|---|---|
| | | N | % | N | % | N | % | |
| Subjective health | Very good | 1,155 | 6.9 | 567 | 7.1 | 588 | 6.8 | <0.01† |
| | Good | 6,002 | 36.1 | 2,769 | 34.5 | 3,233 | 37.5 | |
| | Relatively good | 6,308 | 37.9 | 3,032 | 37.8 | 3,276 | 38.0 | |
| | Relatively bad | 2,405 | 14.5 | 1,219 | 15.2 | 1,186 | 13.8 | |
| | Bad | 616 | 3.7 | 340 | 4.2 | 276 | 3.2 | |
| | Very bad | 156 | 0.9 | 95 | 1.2 | 61 | 0.7 | |
| Numerical variable | | Mean | SD | Mean | SD | Mean | SD | *p* |
| BMI (kg/m$^2$) | | 22.13 | 3.59 | 23.22 | 3.51 | 21.11 | 3.35 | <0.01† |
| Days of exercise per week | | 2.17 | 2.38 | 1.99 | 2.29 | 2.07 | 2.34 | <0.01† |
| PHQ-9 | | 5.02 | 5.30 | 4.78 | 5.37 | 5.22 | 5.22 | <0.01† |
| GAD-7 | | 3.27 | 4.39 | 3.09 | 4.40 | 3.44 | 4.37 | <0.01† |

*10,000 yen≒110–130 USD

† p<0.05

¶ Disease due to which the participant was prohibited by a doctor from exercising, or disease or injury which caused major difficulties walking (e.g., rheumatoid arthritis or bone fracture)

did not change throughout the study period. There was no apparent difference in this trend between males and females. Of the people who reported that they did not exercise at baseline (4,624), 806 (17.4%) reported they had begun to exercise after a year (at the fifth questionnaire). In contrast, 862 (12.6%) of those who reported exercising at baseline (6,841) stopped exercising over the same period. The standard deviations in both males and females increased slightly with time.

To analyse the factors associated with changes in exercise habit, linear regression was conducted using the change in exercise days as the outcome variable. In the early phase of the pandemic (Table 3, left column), a decreased exercise habit was associated with a high income

**Table 2. Dropout rate of participants in the following questionnaires by sex and age group.**

| Sex | Age group | Jan-21 | | Apr-21 | | Jul-21 | | Oct-21 | |
|---|---|---|---|---|---|---|---|---|---|
| | | N | Dropout (%) | N | Dropout (%) | N | Dropout (%) | N | Dropout (%) |
| Male | ≤30 | 643 | 40.5 | 539 | 50.1 | 440 | 59.3 | 347 | 67.9 |
| | 30–39 | 676 | 28.0 | 633 | 32.6 | 572 | 39.1 | 522 | 44.4 |
| | 40–49 | 1,380 | 14.2 | 1,385 | 13.9 | 1,280 | 20.4 | 1,249 | 22.4 |
| | 50–59 | 1,586 | 9.4 | 1,543 | 11.8 | 1,465 | 16.3 | 1,451 | 17.1 |
| | 60–69 | 1,617 | 9.3 | 1,601 | 10.2 | 1,520 | 14.8 | 1,525 | 14.5 |
| | 70–74 | 783 | 9.0 | 794 | 7.7 | 750 | 12.8 | 745 | 13.4 |
| | Total | 6,685 | 16.7 | 6,495 | 19.0 | 6,027 | 24.9 | 5,839 | 27.2 |
| Female | | N | Dropout (%) | N | Dropout (%) | N | Dropout (%) | N | Dropout (%) |
| | ≤30 | 980 | 43.4 | 874 | 49.5 | 629 | 63.7 | 532 | 69.3 |
| | 30–39 | 786 | 29.4 | 754 | 32.3 | 655 | 41.2 | 602 | 46.0 |
| | 40–49 | 1,325 | 19.6 | 1,292 | 21.6 | 1,218 | 26.0 | 1,188 | 27.9 |
| | 50–59 | 1,248 | 20.3 | 1,254 | 19.9 | 1,178 | 24.8 | 1,147 | 26.8 |
| | 60–69 | 1,408 | 21.6 | 1,449 | 19.4 | 1,324 | 26.3 | 1,230 | 31.6 |
| | 70–74 | 611 | 20.0 | 659 | 13.7 | 610 | 20.2 | 553 | 27.6 |
| | Total | 6,358 | 26.2 | 6,282 | 27.1 | 5,614 | 34.9 | 5,252 | 39.1 |

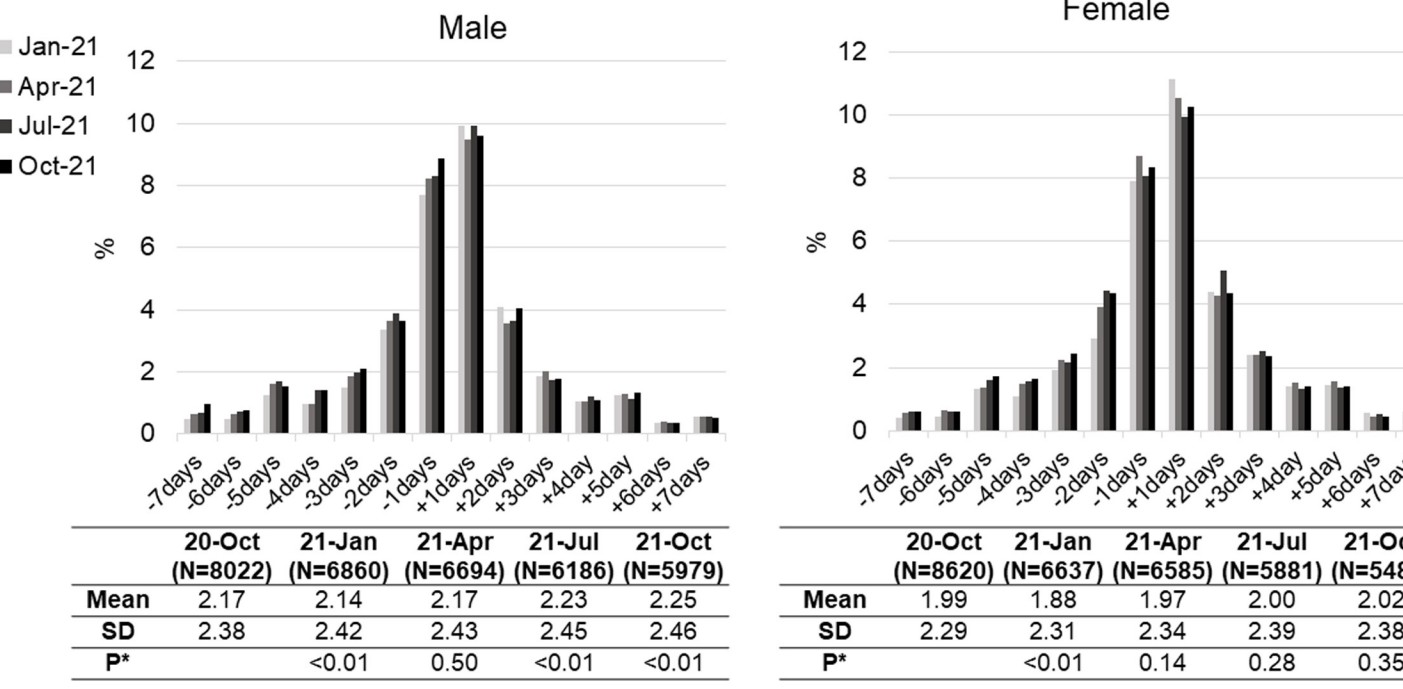

**Fig 1. Change in exercise habit (days per week) compared with the first questionnaire (October 2020).** Only those who changed the habit are included. * p-values of a paired-t test comparing exercise days in each phase with those in October 2020.

(> 10 million yen per year, equivalent to about 100,000 US dollars per year) in both sexes (males, -0.25 days [95% confidence interval -0.44, -0.07]; females, -0.32 days, [-0.51, -0.13]). Elderly females were also associated with decreased exercise days. Having a regular exercise habit at baseline was associated with an increased exercise habit in both sexes (males, 1.08 days [0.97, 1.19]; females, 1.28 days [1.17, 1.39]). An increased exercise habit in women was associated positively with PHQ-9 (0.02 (0.01 to 0.04) and negatively with GAD-7 (-0.03 [-0.05, -0.01]), although the size of this correlation was small.

In the later phase (Table 3, right column), the diagnosis of a SARS-CoV-2 infection was associated with significantly fewer exercise days in males (-0.89 [-1.36, -0.43]). In both sexes, a regular exercise habit at baseline remained associated with increased exercise days (males, 0.13 [0.02, 0.25]; females, 0.20 [0.08, 0.31]). Mental and physical status had no significant association with the changes in exercise habit in the later phase.

## Change in BMI and proportion of obesity/overweight

Another impact caused by the pandemic might be an increase in body weight. The changes in BMI between October 2020 and October 2021 are plotted in S1 Fig. As some people showed a decrease in BMI, just calculating mean BMI may not accurately reflect overweight status. Therefore, the proportion of obesity/overweight and newly developed overweight status as well as mean BMI at each time period are shown in Table 4.

Interestingly, the proportion of obesity appeared to decrease slightly over time, while the proportion of overweight status and mean BMI increased in both sexes. The standard deviation for BMI also increased over time in females. In addition, the proportion of newly developed obesity in males also increased during the first four questionnaires. The increase in BMI and proportion of overweight status was marked in the early phase (mean BMI from 23.22 to

**Table 3. Factors in the early phase that associated with a change in exercise habit in the early and late phase.**

| | | Early phase (January 2021) | | | | | | Late phase (October 2021) | | | | | |
| --- | --- | --- | --- | --- | --- | --- | --- | --- | --- | --- | --- | --- | --- |
| | | Male | | | Female | | | Male | | | Female | | |
| | | Coeff | 95%CI | p | Coeff | 95%CI | p | Coeff | 95%CI | p | Coeff | 95%CI | p |
| BMI (kg/m$^2$) | | 0.00 | -0.01, 0.02 | 0.93 | 0.01 | 0.00, 0.03 | 0.11 | 0.01 | -0.01, 0.03 | 0.20 | -0.01 | -0.03, 0.00 | 0.10 |
| Past diagnosis of COVID-19 | | -0.20 | -0.81, 0.40 | 0.51 | 0.65 | -0.19, 1.49 | 0.13 | -0.89 | -1.36, -0.43 | <0.01[†] | -0.46 | -1.17, 0.26 | 0.21 |
| Age group (yr) | < = 30 | 0 (Reference) | | | 0 (Reference) | | | 0 (Reference) | | | 0 (Reference) | | |
| | 31–40 | 0.01 | -0.21, 0.23 | 0.91 | 0.06 | -0.12, 0.25 | 0.49 | 0.25 | -0.04, 0.54 | 0.09 | 0.09 | -0.17, 0.35 | 0.49 |
| | 41–50 | 0.11 | -0.09, 0.31 | 0.28 | -0.17 | -0.34, -0.01 | 0.04[†] | 0.12 | -0.14, 0.38 | 0.38 | 0.04 | -0.20, 0.27 | 0.75 |
| | 51–60 | 0.12 | -0.09, 0.32 | 0.26 | -0.20 | -0.38, -0.03 | 0.02[†] | 0.24 | -0.02, 0.51 | 0.07 | 0.03 | -0.21, 0.27 | 0.82 |
| | 61–70 | 0.03 | -0.18, 0.24 | 0.78 | -0.26 | -0.45, -0.08 | 0.01[†] | 0.22 | -0.05, 0.50 | 0.12 | -0.01 | -0.26, 0.23 | 0.91 |
| | 71–80 | -0.11 | -0.35, 0.14 | 0.40 | -0.40 | -0.63, -0.17 | <0.01[†] | 0.10 | -0.20, 0.41 | 0.51 | -0.03 | -0.31, 0.26 | 0.84 |
| Income (10,000 yen/year) | <300 | 0 (Reference) | | | 0 (Reference) | | | 0 (Reference) | | | 0 (Reference) | | |
| | 300–500 | -0.02 | -0.16, 0.12 | 0.82 | -0.06 | -0.20, 0.07 | 0.38 | -0.06 | -0.21, 0.10 | 0.47 | 0.00 | -0.16, 0.16 | 0.99 |
| | 500–700 | -0.03 | -0.19, 0.13 | 0.72 | -0.06 | -0.22, 0.09 | 0.41 | -0.02 | -0.19, 0.15 | 0.83 | 0.06 | -0.12, 0.23 | 0.54 |
| | 700–1000 | -0.14 | -0.31, 0.02 | 0.09 | 0.02 | -0.14, 0.19 | 0.77 | 0.02 | -0.16, 0.20 | 0.83 | -0.10 | -0.29, 0.09 | 0.29 |
| | >1000 | -0.25 | -0.44, -0.07 | 0.01[†] | -0.32 | -0.51, -0.13 | <0.01[†] | -0.09 | -0.29, 0.10 | 0.35 | 0.12 | -0.10, 0.34 | 0.28 |
| Married | | -0.11 | -0.22, 0.01 | 0.08 | 0.12 | 0.01, 0.24 | 0.03[†] | 0.02 | -0.10, 0.15 | 0.71 | 0.01 | -0.12, 0.14 | 0.85 |
| Income | No change | 0 (Reference) | | | 0 (Reference) | | | 0 (Reference) | | | 0 (Reference) | | |
| | Increase | -0.06 | -0.17, 0.05 | 0.31 | -0.05 | -0.16, 0.06 | 0.37 | -0.01 | -0.13, 0.11 | 0.92 | 0.03 | -0.10, 0.16 | 0.63 |
| | Decrease | 0.30 | 0.07, 0.53 | 0.01[†] | -0.28 | -0.52, -0.03 | 0.03[†] | -0.18 | -0.44, 0.08 | 0.18 | 0.16 | -0.14, 0.46 | 0.30 |
| Lifestyle | Avoid poorly ventilated places | -0.04 | -0.20, 0.12 | 0.62 | -0.13 | -0.32, 0.07 | 0.21 | 0.09 | -0.11, 0.29 | 0.39 | 0.03 | -0.24, 0.31 | 0.81 |
| | Avoid crowded places | 0.07 | -0.10, 0.25 | 0.42 | 0.03 | -0.16, 0.22 | 0.76 | -0.17 | -0.40, 0.05 | 0.13 | 0.00 | -0.28, 0.28 | 0.99 |
| | Avoid talking at close distances | -0.09 | -0.24, 0.06 | 0.26 | -0.05 | -0.21, 0.10 | 0.49 | 0.12 | -0.08, 0.31 | 0.25 | 0.00 | -0.22, 0.22 | 0.98 |
| | Wear a mask | 0.06 | -0.21, 0.32 | 0.67 | 0.16 | -0.34, 0.65 | 0.54 | 0.19 | -0.13, 0.52 | 0.25 | 0.02 | -0.63, 0.67 | 0.95 |
| | Wash hands | -0.15 | -0.40, 0.10 | 0.25 | -0.16 | -0.56, 0.24 | 0.43 | 0.06 | -0.20, 0.32 | 0.66 | -0.06 | -0.51, 0.40 | 0.80 |
| | Change clothes frequently | 0.02 | -0.13, 0.17 | 0.79 | 0.00 | -0.13, 0.13 | 0.98 | 0.07 | -0.09, 0.22 | 0.39 | -0.13 | -0.27, 0.01 | 0.08 |
| | Gargle | 0.03 | -0.08, 0.14 | 0.58 | -0.04 | -0.16, 0.08 | 0.51 | -0.07 | -0.20, 0.05 | 0.26 | 0.05 | -0.08, 0.19 | 0.44 |
| | Disinfect belongings | -0.09 | -0.23, 0.05 | 0.21 | -0.10 | -0.21, 0.02 | 0.12 | 0.03 | -0.12, 0.18 | 0.66 | -0.04 | -0.17, 0.10 | 0.60 |
| | Keep distance from others | -0.01 | -0.16, 0.13 | 0.85 | 0.02 | -0.14, 0.19 | 0.78 | -0.13 | -0.31, 0.05 | 0.15 | -0.05 | -0.26, 0.17 | 0.65 |
| | Refrain from seeing a doctor | 0.01 | -0.10, 0.12 | 0.84 | -0.04 | -0.15, 0.07 | 0.51 | -0.02 | -0.15, 0.10 | 0.73 | 0.04 | -0.08, 0.17 | 0.50 |
| | Refrain from going out | 0.09 | -0.03, 0.20 | 0.13 | 0.16 | 0.05, 0.27 | 0.01[†] | 0.02 | -0.10, 0.15 | 0.71 | 0.00 | -0.14, 0.14 | 0.99 |
| | Exercise regularly | 1.08 | 0.97, 1.19 | <0.01[†] | 1.28 | 1.17, 1.39 | <0.01[†] | 0.13 | 0.02, 0.25 | 0.02[†] | 0.20 | 0.08, 0.31 | <0.01[†] |
| Subjective health | Very good | 0 (Reference) | | | 0 (Reference) | | | 0 (Reference) | | | 0 (Reference) | | |
| | Good | 0.07 | -0.14, 0.28 | 0.53 | 0.14 | -0.07, 0.35 | 0.19 | 0.03 | -0.21, 0.27 | 0.81 | -0.08 | -0.35, 0.18 | 0.54 |
| | Relatively good | 0.26 | 0.04, 0.47 | 0.02[†] | 0.12 | -0.09, 0.33 | 0.26 | -0.09 | -0.33, 0.15 | 0.47 | -0.21 | -0.47, 0.06 | 0.13 |
| | Relatively bad | 0.26 | 0.02, 0.51 | 0.03[†] | 0.24 | -0.01, 0.48 | 0.06 | -0.13 | -0.40, 0.14 | 0.34 | -0.26 | -0.56, 0.04 | 0.09 |
| | Bad | 0.41 | 0.08, 0.74 | 0.01[†] | 0.21 | -0.15, 0.57 | 0.25 | -0.12 | -0.48, 0.24 | 0.51 | -0.40 | -0.82, 0.01 | 0.06 |
| | Very bad | 0.15 | -0.39, 0.69 | 0.58 | 0.20 | -0.40, 0.81 | 0.51 | -0.16 | -0.66, 0.35 | 0.54 | -0.51 | -1.12, 0.11 | 0.11 |

(*Continued*)

**Table 3.** (Continued)

| | | | | | | | | | | | | |
|---|---|---|---|---|---|---|---|---|---|---|---|---|
| PHQ-9 | 0.01 | -0.01, 0.03 | 0.34 | 0.02 | 0.01, 0.04 | 0.01† | 0.02 | 0.00, 0.04 | 0.11 | 0.00 | -0.02, 0.02 | 0.98 |
| GAD-7 | -0.01 | -0.03, 0.01 | 0.31 | -0.03 | -0.05, -0.01 | 0.01† | -0.03 | -0.05, 0.00 | 0.02† | 0.00 | -0.02, 0.03 | 0.75 |

Controlled for pre-existing conditions.

*10,000yen≒100~130 USD

† p<0.05

23.41 in males and from 21.13 to 21.20 in females; proportion of overweight status from 22.2% to 26.6% in males and from 9.3% to 10.8% in females). In the later phase, the change became less marked, but remained statistically significant in females.

## Risk factors for developing overweight status

As the proportion of obesity was too small to conduct further analysis, factors associated with the development of overweight status were determined by multiple logistic regression. In the early phase (January 2021) of the pandemic, the risk of developing overweight status was significantly higher in middle-aged males (31–70 years old) and elderly females (71–80 years old) (Table 5, left column). Males who were married were more likely to become overweight (odds ratio [OR] 1.61 [1.20, 2.16]), although this change was not observed in married females. On the other hand, females who frequently changed their clothes to prevent a COVID-19 infection (OR 1.68 [1.69, 2.60]) or those with a very bad subjective health condition were more likely to develop overweight status. An increase in income was also associated with the development of overweight status in females (OR 1.54 [1.08, 2.20]), but not in males (OR 0.78 [0.60, 1.01]).

In the late phase (October 2021) (Table 5, right column), males in the age group of 41–50 yr constantly showed a higher risk of becoming overweight (OR 2.35 [1.16, 4.73]). Interestingly, males who were diagnosed with a SARS-CoV-2 infection were also more likely to develop overweight status (OR 2.57 [1.18, 5.60]). Avoiding talking at close distances (OR 0.62 [0.41, 0.94]) and keeping distance from others (OR 0.64 [0.44, 0.94]) were also associated significantly with a lower risk of developing overweight status in males. In females, avoiding poorly ventilated place was associated with a lower risk of becoming overweight (OR 0.34 [0.15, 0.78]).

**Table 4. Fluctuations in body mass index (BMI), proportion of obesity/overweight, and proportion of newly-developed obesity/overweight.** For BMI, the values at each time point were compared with those at baseline (October 2020) using the paired t-test.

| | | Oct-20 | | Jan-21 | | | Apr-21 | | | Jul-21 | | | Oct-21 | | |
|---|---|---|---|---|---|---|---|---|---|---|---|---|---|---|---|
| | | Mean | SD | Mean | SD | p | Mean | SD | p | Mean | SD | p | Mean | SD | P |
| BMI (kg/m$^2$) | Male | 23.22 | 3.54 | 23.41 | 3.99 | <0.01 | 23.39 | 3.79 | <0.01 | 23.34 | 3.78 | 0.07 | 23.34 | 3.74 | 0.08 |
| | Female | 21.13 | 3.49 | 21.20 | 3.91 | <0.01 | 21.18 | 3.79 | <0.01 | 21.14 | 3.81 | <0.01 | 21.16 | 4.03 | <0.01 |
| Obesity (%) | Male | 4.5 | | 4.1 | | | 4.3 | | | 4.2 | | | 4.0 | | |
| | Female | 2.4 | | 2.3 | | | 1.9 | | | 1.9 | | | 2.0 | | |
| Newly developed obesity (%) | Male | Base | | 0.4 | | | 0.5 | | | 0.6 | | | 0.5 | | |
| | Female | Base | | 0.4 | | | 0.3 | | | 0.3 | | | 0.4 | | |
| Overweight status (%) | Male | 22.2 | | 26.6 | | | 26.2 | | | 25.9 | | | 26.1 | | |
| | Female | 9.3 | | 10.8 | | | 10.7 | | | 10.6 | | | 10.5 | | |
| Newly developed overweight status (%) | Male | Base | | 6.9 | | | 6.8 | | | 6.9 | | | 7.2 | | |
| | Female | Base | | 2.7 | | | 2.9 | | | 3.0 | | | 2.9 | | |

**Table 5. Odds ratios for newly developed overweight status in the early and late phases of the pandemic, grouped by sex.** Controlled for pre-existing conditions.

| | | Early phase (January 2021) | | | | | | Late phase (October 2022) | | | | | |
| | | Male | | | Female | | | Male | | | Female | | |
| | | OR | 95%CI | p | OR | 95%CI | p | OR | 95%CI | p | OR | 95%CI | P |
|---|---|---|---|---|---|---|---|---|---|---|---|---|---|
| **Past diagnosis of COVID-19** | | **0.63** | **0.14, 2.84** | **0.55** | NC | | | **2.57** | **1.18, 5.60** | **0.02**[†] | **1.55** | **0.20, 12.13** | **0.68** |
| Age group (yr) | <= 30 | 1 (Reference) | | | 1 (Reference) | | | 1 (Reference) | | | 1 (Reference) | | |
| | 31–40 | 2.72 | 1.38, 5.35 | <0.01[†] | 0.93 | 0.42, 2.04 | 0.86 | 1.62 | 0.75, 3.52 | 0.22 | 0.57 | 0.21, 1.56 | 0.27 |
| | 41–50 | 2.79 | 1.48, 5.29 | <0.01[†] | 0.99 | 0.50, 1.99 | 0.99 | 2.35 | 1.16, 4.73 | 0.02[†] | 0.66 | 0.28, 1.53 | 0.33 |
| | 51–60 | 2.65 | 1.39, 5.06 | <0.01[†] | 1.28 | 0.64, 2.55 | 0.49 | 1.93 | 0.95, 3.94 | 0.07 | 0.88 | 0.39, 2.03 | 0.77 |
| | 61–70 | 2.41 | 1.24, 4.68 | <0.01[†] | 1.70 | 0.86, 3.37 | 0.13 | 1.75 | 0.84, 3.64 | 0.14 | 1.02 | 0.44, 2.36 | 0.96 |
| | 71–80 | 1.79 | 0.87, 3.69 | 0.11 | 2.30 | 1.07, 4.94 | 0.03[†] | 1.22 | 0.55, 2.72 | 0.63 | 1.59 | 0.65, 3.90 | 0.31 |
| Income (yen/year)* | <300 | 1 (Reference) | | | 1 (Reference) | | | 1 (Reference) | | | 1 (Reference) | | |
| | 300–500 | 1.06 | 0.76, 1.48 | 0.73 | 1.04 | 0.66, 1.64 | 0.86 | 1.26 | 0.88, 1.80 | 0.22 | 0.77 | 0.45, 1.30 | 0.32 |
| | 500–700 | 0.92 | 0.64, 1.34 | 0.68 | 1.03 | 0.61, 1.75 | 0.91 | 1.00 | 0.67, 1.51 | 0.98 | 0.81 | 0.44, 1.49 | 0.50 |
| | 700–1000 | 1.17 | 0.81, 1.70 | 0.40 | 0.81 | 0.44, 1.49 | 0.49 | 1.04 | 0.68, 1.58 | 0.86 | 0.95 | 0.50, 1.78 | 0.86 |
| | >1000 | 0.95 | 0.62, 1.45 | 0.80 | 1.11 | 0.57, 2.17 | 0.75 | 1.37 | 0.89, 2.11 | 0.16 | 0.99 | 0.49, 2.00 | 0.98 |
| Married | | 1.61 | 1.20, 2.16 | <0.01[†] | 1.42 | 0.94, 2.14 | 0.10 | 1.08 | 0.80, 1.46 | 0.61 | 1.47 | 0.92, 2.35 | 0.11 |
| Income change | No change | 1 (Reference) | | | 1 (Reference) | | | 1 (Reference) | | | 1 (Reference) | | |
| | Increase | 0.78 | 0.60, 1.01 | 0.06 | 1.54 | 1.08, 2.20 | 0.02[†] | 0.89 | 0.63, 1.24 | 0.49 | 0.90 | 0.52, 1.54 | 0.70 |
| | Decrease | 1.02 | 0.60, 1.72 | 0.94 | 1.21 | 0.48, 3.07 | 0.69 | 1.67 | 0.86, 3.23 | 0.13 | 2.01 | 0.70, 5.83 | 0.20 |
| Lifestyle | Avoid poorly ventilated places | 1.14 | 0.79, 1.65 | 0.49 | 0.60 | 0.31, 1.17 | 0.13 | 1.63 | 1.00, 2.64 | 0.05 | 0.34 | 0.15, 0.78 | 0.01[†] |
| | Avoid places where many people gather | 0.92 | 0.62, 1.37 | 0.68 | 0.87 | 0.45, 1.70 | 0.69 | 1.18 | 0.71, 1.94 | 0.53 | 1.48 | 0.54, 4.02 | 0.44 |
| | Avoid talking at close distances | 0.83 | 0.60, 1.16 | 0.27 | 1.29 | 0.74, 2.26 | 0.37 | 0.62 | 0.41, 0.94 | 0.02[†] | 1.34 | 0.62, 2.87 | 0.46 |
| | Wear a mask | 1.17 | 0.64, 2.16 | 0.61 | 2.22 | 0.27, 18.53 | 0.46 | 1.61 | 0.74, 3.51 | 0.23 | 0.79 | 0.08, 7.33 | 0.83 |
| | Wash hands | 0.85 | 0.49, 1.46 | 0.55 | 0.87 | 0.24, 3.18 | 0.84 | 0.72 | 0.41, 1.26 | 0.24 | 2.39 | 0.27, 21.25 | 0.43 |
| | Change clothes frequently | 0.97 | 0.69, 1.36 | 0.86 | 1.68 | 1.09, 2.60 | 0.02[†] | 1.01 | 0.71, 1.43 | 0.97 | 1.43 | 0.89, 2.30 | 0.14 |
| | Gargle | 0.84 | 0.65, 1.08 | 0.17 | 0.92 | 0.61, 1.37 | 0.67 | 1.03 | 0.77, 1.36 | 0.86 | 0.89 | 0.56, 1.40 | 0.60 |
| | Disinfect belongings | 1.14 | 0.84, 1.56 | 0.39 | 0.80 | 0.52, 1.23 | 0.31 | 1.06 | 0.75, 1.48 | 0.75 | 0.72 | 0.44, 1.16 | 0.18 |
| | Keep distance from others | 0.88 | 0.63, 1.23 | 0.46 | 1.00 | 0.56, 1.78 | 1.00 | 0.64 | 0.44, 0.94 | 0.02[†] | 1.01 | 0.47, 2.15 | 0.98 |
| | Refrain from seeing a doctor | 0.86 | 0.67, 1.11 | 0.24 | 0.67 | 0.46, 0.99 | 0.04[†] | 1.14 | 0.86, 1.52 | 0.36 | 1.31 | 0.85, 2.02 | 0.22 |
| | Refrain from going out | 1.21 | 0.94, 1.56 | 0.15 | 1.30 | 0.87, 1.95 | 0.20 | 1.02 | 0.77, 1.36 | 0.88 | 0.93 | 0.58, 1.48 | 0.76 |
| | Exercise regularly | 0.97 | 0.76, 1.22 | 0.78 | 0.77 | 0.54, 1.10 | 0.15 | 0.95 | 0.73, 1.23 | 0.70 | 1.07 | 0.71, 1.61 | 0.74 |
| Subjective health | Very good | 1 (Reference) | | | 1 (Reference) | | | 1 (Reference) | | | 1 (Reference) | | |
| | Good | 1.30 | 0.76, 2.24 | 0.34 | 0.69 | 0.33, 1.46 | 0.33 | 1.16 | 0.67, 2.01 | 0.61 | 0.53 | 0.23, 1.24 | 0.14 |
| | Relatively good | 1.35 | 0.78, 2.34 | 0.28 | 0.89 | 0.42, 1.86 | 0.75 | 0.98 | 0.56, 1.73 | 0.95 | 1.00 | 0.44, 2.30 | 1.00 |
| | Relatively bad | 1.44 | 0.78, 2.64 | 0.24 | 0.96 | 0.41, 2.23 | 0.92 | 1.03 | 0.55, 1.94 | 0.92 | 0.97 | 0.38, 2.52 | 0.96 |
| | Bad | 0.78 | 0.33, 1.86 | 0.57 | 0.93 | 0.27, 3.15 | 0.91 | 1.14 | 0.50, 2.62 | 0.75 | 0.47 | 0.09, 2.55 | 0.38 |
| | Very bad | 1.61 | 0.46, 5.59 | 0.46 | 4.79 | 1.16, 19.7 | 0.03[†] | 1.53 | 0.51, 4.54 | 0.45 | 0.82 | 0.08, 8.14 | 0.87 |
| PHQ-9 | | 1.02 | 0.98, 1.07 | 0.33 | 1.01 | 0.95, 1.08 | 0.68 | 1.01 | 0.96, 1.05 | 0.80 | 0.99 | 0.92, 1.06 | 0.77 |
| GAD-7 | | 1.00 | 0.95, 1.05 | 0.86 | 0.99 | 0.93, 1.06 | 0.84 | 0.99 | 0.94, 1.05 | 0.82 | 1.03 | 0.95, 1.12 | 0.51 |

OR, odds ratio; CI, confidence interval

*10,000 yen≒110-130USD

† $p < 0.05$

## Long-term impact of the conditions in the early phase of the pandemic on the onset of overweight status

Assuming that overweight status in the late phase (October 2021) was affected by factors in the early phase (October 2020), further analysis was carried out on the association between being

overweight in the late phase and lifestyle factors in the early phase as a sensitivity analysis (S2 Table).

In males, infection with the SARS-CoV-2 in the early phase correlated significantly with the development of overweight status in the late phase (OR 3.01 [1.27, 7.13]), while those who experienced a decrease in income showed a lower risk (OR 0.73 [0.54, 0.97]). On the other hand, females whose income decreased in the early phase were more likely to become overweight in the late phase (OR 1.75 [1.18, 2.61]). Although not statistically significant, there was a trend that females who had a worse subjective health score in the early phase were more likely to have a higher risk of prolonged overweight status. Exercise habit was not associated with the risk of developing overweight status in any of the analyses.

## Discussion

This study included novel, nationwide, longitudinal research on exercise habits and overweight risks in Japan during the COVID-19 pandemic. The study showed a trend of a decrease in exercise habit and increase in overweight status among a group of the population. This suggested the COVID-19 pandemic had a strong negative impact associated with a restriction of social activities. However, our research also showed that the proportion of obesity status actually decreased during the pandemic period, suggesting the impact was heterogeneous. This finding is consistent with those of previous studies [16, 17]. This may mean that targeted intervention, but not general intervention, may be required to prevent the impact of the disaster on obesity-related health outcomes. In addition, our research showed that the factors that associate with immobility and overweight status were different. Therefore intervention to prevent these two health problems might be considered independently.

Previous reports suggest that prolonged evacuation may increase the risk of chronic conditions including obesity, presumably due to increased mental stress and poor access to healthcare services [25, 26]. Our research suggests that depression and anxiety had limited impact on the health problems, suggesting there might be other cause of health deterioration during the pandemic.

### Older females as a vulnerable population in the COVID-19 pandemic

Importantly, elderly females appeared to be at higher risk for both immobility and overweight status in the early phase of the pandemic. These risks also correlated with worse subjective health in females. These results suggest that this trend might be due partly to fear of COVID-19, which has been reported to be higher in females than in males [27]. In addition, the elderly population were more vulnerable to biased reports by mass media [28] and the current infodemic. Therefore, it is possible that the infodemic and other biased information exacerbated the fear elderly females had of COVID-19. This fear may be decreased by fact-checking information [27]. Indeed, in other disasters such as the Fukushima nuclear accident, public communication through the Fukushima health management surveys was effective for reducing anxiety among the residents [29]. Therefore, in future disasters, appropriate intervention in the acute phase may need to include providing the population with scientific-based information as well as information about self-management and psychological first aid targeting the elderly population.

### Bipolarization of the exercise habit

Our study also showed that people who already had an exercise habit were more likely to increase their exercise. Therefore, improving this pre-condition by installing exercise habits before the pandemic in high-risk groups might be another strategy for disaster preparation.

Interestingly, our study showed that a high income (>10 million yen per year) was associated significantly with decreased exercise habits. This may mean that people engaged in administrative work or work with greater responsibility were overwhelmed by their duty during the pandemic, leading to a decrease in their exercise habit. This may also explain why males who experienced decreased income were more likely to increase their exercise habit. In other words, workload and exercise times were a trade-off in males.

On the other hand, females who experienced decreased income were more likely to also decrease their exercise, possibly because those who left their jobs did so due to increased housework [30] or those who started part-time jobs got less salary with longer worktime. Another possible reason is that female whose income decreased were more likely to become depressive. Further research is required to elucidate the reasons why exercise times in females were not a trade-off for a reduction in income.

## Concern about the impact of overweight status on long-term health conditions

In addition to immobility, obesity is one of the major concerns after a huge disaster, especially among evacuees [6, 31]. Lock-down and keeping social distance may have caused the similar effects to evacuation on the public. Indeed, the present research study revealed that about 6% of non-obese males and 3% of non-obese females became overweight during the period of the pandemic. As there was a group of people whose BMI decreased, the net increase in the proportion of overweight was about 4% in males and 1% in females. Above all, middle-aged males were at higher risk of becoming overweight in both the early and late phase of the pandemic. Considering that an increased BMI in middle-age causes loss of life expectancy by 5–13 years [32], this indirect impact of the pandemic should not be ignored. Intervention in the high-risk population is therefore essential to prevent the impact of a disaster on overweight status.

**Risk of overweight among males.**   For males, the diagnosis of COVID-19 was associated significantly with the development of overweight status. This association can be interpreted in several ways. One scenario is that COVID-19 infection may have led to overweight status. As the diagnosis also correlated with a decrease in exercise habits, this increase in overweight status may have been due to a lack of exercise. However, there was no significant difference in exercise days between those who were diagnosed with a SARS-CoV-2 infection and those who were not (Average days of exercise per week in those who were infected and those who were not were 2.30 days and 2.00 days in the early phase (p = 0.16 by t-test) and 2.43 days and 2.13 days in the late phase(p = 0.13)). Another possible reason is that post-COVID syndromes such as post-traumatic disorders, depression, and chronic fatigue may lead to inactivity, thereby increasing the risk of becoming overweight [14]. Some experts consider rehabilitation in the recovery phase of COVID-19 should include not only respiratory and cardiovascular rehabilitation but also muscle training and psychological support [33]. Such interventions may need to be applied for those whose symptoms were less severe. However, to date there are no guidelines regarding interventions for patients who were not hospitalised. An effort to reduce the indirect and prolonged health impacts caused by the SARS-CoV-2 pandemic may need to target this population.

Another scenario is that the development of overweight status has led to increase in the risk of symptomatic SARS-CoV-2 infection. As overweight status and obesity is a risk factor of developing severer symptoms, people in overweight status may have been at higher risks of being diagnosed as SARS-CoV-2 infection. To clarify the causal relationships, further research is required such as long-term follow-up of the infected people.

**Risk of overweight status among females.** Our research also revealed elderly females were at higher risk of developing overweight status in the early phase compared with the other age groups. A previous study reported homemakers were more likely to gain body weight [15], which was consistent with our findings. The factors causing overweight status in elderly people include a decrease in time spent for outings due to the geographically isolated conditions of temporary housing [34] and prolonged post-traumatic stress disorders (PTSD) [35]. During the COVID-19 pandemic, people stayed at home for a longer time, which might have caused similar conditions to long-term evacuation, such as less outings and higher mental stress. Another possible reason is change in eating habit. If people try to go out as seldom as possible, they may buy more preserved food and less fresh fruits and vegetables, which may affect body weight. However, older age confounds with a variety of socio-economic and mental status. For example, elderly people living on pensions or those with dementia may be more likely to be at poorer mental status. Further surveys on the impact of such factors on health status are required.

Overweight status in elderly women may have a marked health impact on society because being overweight in this population group is a significant risk factor for immobility and frailty, which may lead to bone fracture or a bed-ridden state [36, 37]. Therefore, immediate intervention might have been needed to target this group of people in the early phase of the pandemic. For females, the development of overweight status was associated with seemingly excessive reactions against SARS-CoV-2, such as changing clothes frequently. As bad subjective health status was associated with the risk of developing overweight status, anxiety might also have been a risk factor.

To prevent lifestyle diseases, interventions by health professionals are not sufficient. In addition, the health system is often severely compromised in the affected areas due to overwhelming demand, evacuation of healthcare workers [38], diversion of resources, and closure of health facilities [39]. Therefore, self-management such as regular exercise and weight control is a key to disaster mitigation.

## Limitations

This study had several limitations. First, the study relied solely on participant responses and therefore we could not avoid false answers even after excluding those that were apparently controversial. In addition, several important questions that may affect body weight, such as eating habits, specific cause of mental stresses such as increase in housework, are not included in the questionnaire mainly due to lack of finance. The number of questionnaires was also limited to five times from the same reason. Second, although the participants were matched to the national demographic background, dropout rate was different between sexes and age groups. There also remains selection bias of the participants. For example, individuals who could not read Japanese and those who could not use the internet were excluded. In addition, individuals with a history of infection could have more actively sought to participate in our study because of their increased interest in the significance and content of this online survey, causing an upward bias in participation of this type of subject. Indeed, our data showed the proportion of those who had exercise habit of ≥ 2 days per week was higher than the that of The National Health and Nutrition Survey in Japan 2019, which may reflect these selection bias. Third, about one-third of the participants missed some of the data during the survey period. As there were some significant differences between those with missing data and those with complete data (S1 Table), these numbers may have affected the generalizability of our results. Forth, causal relationships cannot be proved by this survey. For example, it is not clear whether newly developed overweight status increased the risk of COVID-19 infection or vice-versa. By

using the factors in the early phase as explanatory variables and newly developed overweight status as an outcome variable, this limitation could be partially overcome. Fifth, there are many potential confounders that were not asked in the questionnaire. For example, decrease in exercise day does not always mean decline in activity -before the pandemic the average commuting hours of Japanese businesspeople was about 50 minutes, which could be substitute for exercise time [40]. Therefore, it is possible that increase in working at home may have decreased overall activities even when the exercise day increased. Finally, the survey did not include that of genetic factors, which may account for 40 to 50% of variability in body weight status [41]. In addition, there might be difference in genetic backgrounds between Japan and other countries, which may limit the generalizability of our findings. To elucidate more detailed causal relationships, further research such as prospective study of physical performance tests and surveys including blood testing is required. However, despite these limitations, our research provided sufficient generalizability compared to other studies because of the broadness of the participants' background.

## Conclusion

This study analysed the impact of the COVID-19 pandemic on exercise habit and the development of overweight status in the Japanese population. Risk factors for these conditions were shown to be different between the sexes. Our results suggest that early intervention for elderly women such as provision of information and mental care, and long-term intervention including physical and mental rehabilitation for people who were infected might have been needed during the pandemic. As most CBRNE disasters cause similar social transformation, intervention to prevent immobility and obesity among the high-risk population should be addressed in future disaster preparation/mitigation plans so that we can prevent distant health impacts associated with a disaster. Further research is still needed to clarify the detailed factors that affect exercise habits and overweight status, such as eating habits, change in the volume of housework, other causes of mental stress and genetic factors that may impact body weight status.

## Supporting information

**S1 Table. Background of the participants who dropped out during the surveillance period.**
(DOCX)

**S2 Table. Impact of the factors on prolonged overweight in the early phase of the pandemic.**
(DOCX)

**S1 Fig. Plot of the body mass index of each participant in October 2020 (horizontal) and October 2021 (vertical).**
(TIF)

## Acknowledgments

We would like to thank Dr Kenzo Denda, Hiramatsu Memorial Hospital for his support in ethical considerations of the study.

## Author Contributions

**Conceptualization:** Sae Ochi, So Mirai, Sora Hashimoto, Yuki Hashimoto, Yoichi Sekizawa.

**Data curation:** Sae Ochi, So Mirai, Yuki Hashimoto, Yoichi Sekizawa.

**Formal analysis:** Sae Ochi, So Mirai, Sora Hashimoto.

**Funding acquisition:** Yoichi Sekizawa.

**Investigation:** Sae Ochi.

**Methodology:** Sae Ochi, Sora Hashimoto, Yuki Hashimoto, Yoichi Sekizawa.

**Project administration:** Yuki Hashimoto, Yoichi Sekizawa.

**Resources:** Yoichi Sekizawa.

**Software:** Sae Ochi, Yoichi Sekizawa.

**Supervision:** So Mirai, Yuki Hashimoto.

**Validation:** Sae Ochi, Sora Hashimoto, Yuki Hashimoto, Yoichi Sekizawa.

**Visualization:** Sae Ochi, Yuki Hashimoto.

**Writing – original draft:** Sae Ochi.

**Writing – review & editing:** So Mirai, Sora Hashimoto, Yuki Hashimoto, Yoichi Sekizawa.

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
