## [Decision Letter · Decision Letter 0]

27 Mar 2023

PGPH-D-23-00212

Impact of the COVID-19 pandemic on exercise habits and overweight in Japan: a nation-wide panel survey

Dear Ochi,

Thank you for submitting your manuscript to PLOS Global Public Health. After careful consideration, we feel that it has merit but does not fully meet PLOS Global Public Health’s publication criteria as it currently stands. Therefore, we invite you to submit a revised version of the manuscript that addresses the points raised during the review process.

We look forward to receiving your revised manuscript.

Kind regards,

Collins Otieno Asweto, PhD

Academic Editor

Journal Requirements:

1. We suggest you thoroughly copyedit your manuscript for language usage, spelling, and grammar. If you do not know anyone who can help you do this, you may wish to consider employing a professional scientific editing service.

2. Please send a completed 'Competing Interests' statement, including any COIs declared by your co-authors. If you have no competing interests to declare, please state "The authors have declared that no competing interests exist". Otherwise please declare all competing interests beginning with the statement "I have read the journal's policy and the authors of this manuscript have the following competing interests:"

3. Please provide a/amend your detailed Financial Disclosure statement. This is published with the article. It must therefore be completed in full sentences and contain the exact wording you wish to be published.

a. Please clarify all sources of funding (financial or material support) for your study. List the grants (with grant number) or organizations (with url) that supported your study, including funding received from your institution. 

b. State the initials, alongside each funding source, of each author to receive each grant.

c. State what role the funders took in the study. If the funders had no role in your study, please state: “The funders had no role in study design, data collection and analysis, decision to publish, or preparation of the manuscript.”

4. Please provide separate figure files in .tif or .eps format.

5. We noticed that you used “data not shown”/"unpublished data" in the manuscript. We do not allow these references, as the PLOS data access policy requires that all data be either published with the manuscript or made available in a publicly accessible database. Please amend the supplementary material to include the referenced data or remove the references.

6. Since your data is not available for proprietary reasons, please explain via email why the data is not available. Please also include the contact information for the third party organization that should be contacted should other researchers want to request access to this data and please include the full citation of where the data can be found. We also request that you verify with us via email that any researcher will be able to obtain the data set in the same manner that the you have obtained it. If you feel you are unwilling or unable to adhere to this policy, please explain your reasons by return email and your exemption request will be escalated to the editor for approval. Your exemption request will be handled independently and will not hold up the peer review process, but will need to be resolved should your manuscript be accepted for publication. One of the Editorial team will be in touch if they require more information.

7. We have noticed that you have uploaded Supporting Information files, but you have not included a list of legends. Please add a full list of legends for your Supporting Information files after the references list.  

Reviewers' comments:

Reviewer's Responses to Questions

**Comments to the Author**

1. Does this manuscript meet PLOS Global Public Health’s publication criteria? Is the manuscript technically sound, and do the data support the conclusions? The manuscript must describe methodologically and ethically rigorous research with conclusions that are appropriately drawn based on the data presented.

Reviewer #1: Partly

Reviewer #2: Yes

2. Has the statistical analysis been performed appropriately and rigorously?

Reviewer #1: I don't know

Reviewer #2: Yes

3. Have the authors made all data underlying the findings in their manuscript fully available (please refer to the Data Availability Statement at the start of the manuscript PDF file)?

Reviewer #1: Yes

Reviewer #2: Yes

4. Is the manuscript presented in an intelligible fashion and written in standard English?

Reviewer #1: Yes

Reviewer #2: Yes

5. Review Comments to the Author

Reviewer #1: Interesting and important topic

Well justified

Methodology is clear

Clear table and presentation of data

Minor Edits:

- Not clear how the COVID-19 pandemic falls under the definition of CBRNE. Perhaps there is no need to put it under this category as it becomes vague and inaccurate.

- make it more clear regarding correlation and not causation (use correlation instead of may cause), and add more on that in the limitations section.

-Add what further studies are recommended to investigate this correlation/causation further - in the conclusion section.

-In discussion : add more on role of possible confounding factors that are missed in this study (e.g. eating habits).

Reviewer #2: Thank you very much. I have enjoyed reading the article. It is always great to have numbers to support the personal observations.

The methodology is clear with the suitable exclusion criteria, proper analysis and presentation of findings.

6. PLOS authors have the option to publish the peer review history of their article (what does this mean?). If published, this will include your full peer review and any attached files.

**Do you want your identity to be public for this peer review?** For information about this choice, including consent withdrawal, please see our Privacy Policy.

Reviewer #1: No

Reviewer #2: No

---

## [Decision Letter · Decision Letter 1]

31 May 2023

PGPH-D-23-00212R1

Impact of the COVID-19 pandemic on exercise habits and overweight status in Japan: a nation-wide panel survey

Dear Sae,

Thank you for submitting your manuscript to PLOS Global Public Health. After careful consideration, we feel that it has merit but does not fully meet PLOS Global Public Health’s publication criteria as it currently stands. Therefore, we invite you to submit a revised version of the manuscript that addresses the points raised during the review process.

Please ensure that your decision is justified on PLOS Global Public Health’s publication criteria and not, for example, on novelty or perceived impact.

We look forward to receiving your revised manuscript.

Kind regards,

Collins Otieno Asweto, PhD

Academic Editor

Journal Requirements:

1. Please submit a copy edited of your manuscript.

Additional Editor Comments (if provided):

Reviewers' comments:

Reviewer's Responses to Questions

**Comments to the Author**

1. If the authors have adequately addressed your comments raised in a previous round of review and you feel that this manuscript is now acceptable for publication, you may indicate that here to bypass the “Comments to the Author” section, enter your conflict of interest statement in the “Confidential to Editor” section, and submit your "Accept" recommendation.

Reviewer #1: All comments have been addressed

Reviewer #3: All comments have been addressed

2. Does this manuscript meet PLOS Global Public Health’s publication criteria? Is the manuscript technically sound, and do the data support the conclusions? The manuscript must describe methodologically and ethically rigorous research with conclusions that are appropriately drawn based on the data presented.

Reviewer #1: Yes

Reviewer #3: Yes

3. Has the statistical analysis been performed appropriately and rigorously?

Reviewer #1: Yes

Reviewer #3: Yes

4. Have the authors made all data underlying the findings in their manuscript fully available (please refer to the Data Availability Statement at the start of the manuscript PDF file)?

Reviewer #1: Yes

Reviewer #3: Yes

5. Is the manuscript presented in an intelligible fashion and written in standard English?

Reviewer #1: (No Response)

Reviewer #3: Yes

6. Review Comments to the Author

Reviewer #1: Comments have been appropriately addressed

Reviewer #3: IMPACT OF THE COVID-19 PANDEMIC ON EXERCISE HABITS AND OVERWEIGHT STATUS IN JAPAN: A NATION-WIDE PANEL SURVEY.

This research by Sae Ochi et al, aimed to analyse the association of the COVID-19 PANDEMIC with lifestyle factors of exercise habits and overweight status in the Japanese population.

The study deployed online nation wide questionnaire which were administered five times between October 2020 and October 2021. The changes were compared between the first questionnaire and the later questionnaire. Analysis of the risk factors for losing exercise habit or becoming overweight were done using multiple regression. The findings suggest that the risks for immobility and overweight are homogeneous.

COMMENTS:

The authors did a fantastic job by painstakingly looking at the health impacts of Covid 19 on lifestyle changes such as exercise habits and overweight weight status of the Japanese population. The results of this study will surely provide adequate insights on the health impacts of the Covid-19 pandemic.

1. The title is adequate and the abstract gave a good summary of what was done with essential information required.

Line 7: under abstract : "compared between the first and later questionnaire": this appear vague, the authors should clarify the term "later questionnaire" specifically for easy reference.

2. Data presented and analyzed in this study were post-covid , a brief background introductory information on the pre-covid lifestyle factors of exercise habits and overweight status among Japanese population would give readers a more robust understanding of the impact of this study. The authors may wish to consider this.

3. MATERIALS AND METHODS:

Line 7 : Online questionnaire were conducted five times.

Why five times? What was the rationale for this? Were the same questions administered five times on the same or different populations? How was the baseline data determined? Were there pre and post data comparison? Which of these represented the baseline/pre and post data collected? The authors may wish to clarify all these for easy comprehension.

4. DEFINITION OF CHANGES IN THE EARLY AND LATE PHASES

Line 3: Reference was made to first, second and fifth questionnaire.

Was there a third and fourth questionnaire?

If the questionnaires were administered five times, an explanation or reference may be included in the study for third and fourth questionnaire.

5. TARGET POPULATION

Line 14 & 15: mentions selection of target population from the database, was there a scientific determination for this selection? How did you arrive at a sample size of 2000 for the study ( line 17). A simple clarification will be fine.

6. STATISTICAL ANALYSIS.

The statistical tools of analysis deployed by the authors were in harmony with the data presented and this produced an incisive outlook of the research aims.

7. In addition to some risk factors mentioned, genetics is another major risk factor in being overweight.

The authors may wish to list out all the possible risk factors for being overweight but clarify that the focus of this study is on the impact of covid-19 pandemic.

8. The massive work done by these authors are very commendable. This research is unique and interesting with a lot of information that will be added to the body of knowledge.

Conclusion: The authors have addressed the concerns raised by the editor and previous reviewer. They may wish to address these fresh concerns I have raised. However, this research is massive and the authors have done a commendable work.

7. PLOS authors have the option to publish the peer review history of their article (what does this mean?). If published, this will include your full peer review and any attached files.

**Do you want your identity to be public for this peer review?** For information about this choice, including consent withdrawal, please see our Privacy Policy.

Reviewer #1: No

Reviewer #3: **Yes: **PRISCILIA UHUANMWEN IMADE

---

## [Decision Letter · Decision Letter 2]

28 Jun 2023

Impact of the COVID-19 pandemic on exercise habits and overweight status in Japan: a nation-wide panel survey

PGPH-D-23-00212R2

Dear Ochi,

We are pleased to inform you that your manuscript 'Impact of the COVID-19 pandemic on exercise habits and overweight status in Japan: a nation-wide panel survey' has been provisionally accepted for publication in PLOS Global Public Health.

Best regards,

Collins Otieno Asweto, PhD

Academic Editor

Reviewer Comments (if any, and for reference):

Reviewer's Responses to Questions

**Comments to the Author**

1. If the authors have adequately addressed your comments raised in a previous round of review and you feel that this manuscript is now acceptable for publication, you may indicate that here to bypass the “Comments to the Author” section, enter your conflict of interest statement in the “Confidential to Editor” section, and submit your "Accept" recommendation.

Reviewer #4: All comments have been addressed

Reviewer #5: All comments have been addressed

2. Does this manuscript meet PLOS Global Public Health’s publication criteria? Is the manuscript technically sound, and do the data support the conclusions? The manuscript must describe methodologically and ethically rigorous research with conclusions that are appropriately drawn based on the data presented.

Reviewer #4: Yes

Reviewer #5: Yes

3. Has the statistical analysis been performed appropriately and rigorously?

Reviewer #4: Yes

Reviewer #5: Yes

4. Have the authors made all data underlying the findings in their manuscript fully available (please refer to the Data Availability Statement at the start of the manuscript PDF file)?

Reviewer #4: Yes

Reviewer #5: Yes

5. Is the manuscript presented in an intelligible fashion and written in standard English?

Reviewer #4: Yes

Reviewer #5: Yes

6. Review Comments to the Author

Reviewer #4: Given the sporadic nature of various disasters, the study provides important information that need to be taken on board as part of disaster preparedness and control.

The information was truly presented in an intelligent and coherent manner.

The methods were detailed and clear.

Results: It is important to format a table 2. As it stands, N could be interpreted as people who also dropped out which is not the case here. Aligning the N and its % is recommended.

Ethics consideration: it is important to highlight how ethical issues like confidentiality were handled

Reviewer #5: Table 2 on the drop out rates needs further explanation. The rates in percentage appear to be disproportionate to the actual numbers (N) shown. Perhaps a few sentences explaining this Table will be in order

7. PLOS authors have the option to publish the peer review history of their article (what does this mean?). If published, this will include your full peer review and any attached files.

**Do you want your identity to be public for this peer review?** For information about this choice, including consent withdrawal, please see our Privacy Policy.

Reviewer #4: No

Reviewer #5: **Yes: **Lakshmi Narasimhan Balaji
